# *Stomoxys* flies (Diptera, Muscidae) are competent vectors of *Trypanosoma evansi*, *Trypanosoma vivax*, and other livestock hemopathogens

Julia W. Muita[1,2], Joel L. Bargul[2], JohnMark O. Makwatta[1], Ernest M. Ngatia[1,2], Simon K. Tawich[1], Daniel K. Masiga[1], Merid N. Getahun[1]*

1 International Centre of Insect Physiology and Ecology (icipe), Nairobi, Kenya, 2 Department of Biochemistry, Jomo Kenyatta University of Agriculture and Technology (JKUAT), Nairobi, Kenya

* mgetahun@icipe.org

## Abstract

*Stomoxys* flies are widely distributed and economically significant vectors of various livestock pathogens of veterinary importance. However, the role of *Stomoxys* spp. in pathogen transmission is poorly understood. Therefore, we studied the feeding patterns of these blood feeders collected from specific locations in Kenya to identify various vertebrate hosts they fed on, the livestock hemopathogens they carried, and to elucidate their role in pathogens transmission. Our findings show that field-collected *Stomoxys* flies carried several pathogens, including *Trypanosoma* spp., *Anaplasma* spp., and *Theileria* spp., which were also detected in the blood of sampled livestock, namely camels and cattle. The findings on blood meal analysis show that *Stomoxys* flies fed on various domestic and wild vertebrate hosts. We further determined whether *Stomoxys* spp. are vectors of hemopathogens they harbored by studying the vector competence of *Stomoxys calcitrans, S. niger niger,* and *S. boueti* species complex, through laboratory and natural experimental *in vivo* studies. We show that in the process of blood feeding *Stomoxys* spp. complexes can transmit *Trypanosoma evansi* (8.3%) and *T. vivax* (30%) to Swiss white mice. In addition, field-collected *Stomoxys* spp. were exposed to healthy mice for blood meal acquisition, and in the process of feeding, they transmitted *Theileria mutans* and *Anaplasma* spp. to Swiss white mice (100% infection in the test mice group). All mice infected with trypanosomes via *Stomoxys* bite died while those infected with *Theileria* and *Anaplasma* species did not, demonstrating the virulence difference between pathogens. The key finding of this study showing the wide distribution, broad feeding host range, plethora of pathogens harbored, and efficient vector competence in spreading multiple pathogens suggests the significant role of *Stomoxys* on pathogen transmission and infection prevalence in livestock.

the Creative Commons Attribution License, which permits unrestricted use, distribution, and reproduction in any medium, provided the original author and source are credited.

**Data availability statement:** All relevant data are in the manuscript and supporting information. All pathogens and Stomoxys sequences data are deposited in the NCBI database NCBI accession numbers presented in S1 File.

**Funding:** This project has received funding from the European Union's Horizon 2020 research and innovation program under grant agreement No: 101000467, the acronym 'COMBAT' (Controlling and Progressively Minimizing the Burden of Animal Trypanosomiasis) to JWM, MNG, STK, ENM. Additionally, this project was funded by the Max Planck Institute for Chemical Ecology-icipe partner group to MNG, ENM. Additional funding from European Commission [HORIZON-CL6-2021-FARM2FORK-01-18: One Health sustainability partnership between EU-AFRICA for food security (NESTLER project: 101060762) to ENM. The authors gratefully acknowledge the financial support for this research by the following organizations and agencies, the Swedish International Development Cooperation Agency (Sida); the Swiss Agency for Development and Cooperation (SDC); the Australian Centre for International Agricultural Research (ACIAR); the Government of Norway; the German Federal Ministry for Economic Cooperation and Development (BMZ); and the Government of the Republic of Kenya. The funders had no role in study design, data collection and analysis, decision to publish, or preparation of the manuscript.

**Competing interests:** The authors have declared that no competing interests exist.

## Author summary

*Stomoxys* are highly adaptable to several ecological settings, including metropolitan areas. In contrast, tsetse flies, the main biological vectors of trypanosomes, have a limited distribution in parks and conservation areas. *Stomoxys* could play a significant role in the spread of animal trypanosomes, among other hemopathogens, in areas with or without tsetse infestation. Although there have been speculations about the potential role of *Stomoxys* in the transmission of various pathogens, there is lack of data to link hemopathogens occurring in both blood meal hosts of *Stomoxys* and in the flies, and further *in vivo* experimental studies to confirm the vector competence of Stomoxyine flies. Here, we explored a host and pathogens network, investigated species diversity at various ecologies, and demonstrated that *Stomoxys* feed on diverse vertebrate hosts and are infected with a plethora of pathogens. We also showed experimentally that they could transmit some of these hemopathogens to mice, for instance, *T. vivax, T. evansi, T. mutans,* and *Anaplasma* spp. with varying infection success rates. *Stomoxys* could play a significant role in transmitting and spreading various hemopathogens of veterinary importance and possibly maintaining their circulation in livestock, which could explain the occurrence of animal trypanosomes in the regions outside the tsetse belts.

## Introduction

*Stomoxys* flies are widely distributed globally [1]. They feed on both blood [2,3] and nectar [4]. These blood feeders pose a significant threat to livestock production worldwide, especially because of their occurrence in wider ecological zones [5]. The economic losses due to livestock infestation by *Stomoxys* flies are substantial, leading to a significant reduction in meat and milk production [6], due to pathogen infestation. These pathogens cause diseases, including; animal trypanosomiasis, Rift Valley fever, African swine fever, lumpy skin disease, and anaplasmosis [3,7–10]. The annual economic losses in the United States of America (USA) alone due to infestation by a single *Stomoxys* species, *S. calcitrans*, a cosmopolitan species commonly known as stable fly, have been estimated to be around US$2.2 billion [11]. The combined economic losses caused by *Stomoxys* flies, including infection, nuisance, and treatment costs outside the USA, are not well documented presently.

*Stomoxys* feed on their vertebrate hosts' blood once or twice a day [12]. Since the host animals respond to protect themselves against the painful blood-feeding fly bites, they (*Stomoxys*) rarely complete blood-feeding on a single animal [5]. In the event of an interrupted blood meal, the fly can restart feeding on another host thus, possibly injecting infected saliva before feeding [5]. However, the pathogen transmission mechanism is not clear. Still, it could vary from biological, as is the case in the transmission of filariasis [13,14], to the mechanical transmission of, for instance, animal trypanosomiasis [5,15,16].

Animal trypanosomiasis caused by *T. vivax* and *T. evansi* is endemic in most countries in sub-Saharan Africa outside the tsetse belts [3,15,17,18], and also in other

regions outside Africa, including Asia [19], Latin America [20], and Europe [21]. *T. evansi* is the causative agent of 'surra', and *T. vivax* that of nagana, animal diseases that are endemic in large swathes of Africa, Asia, and Latin America, and also present in the Canary Islands (Spain) [22]. The establishment, widespread occurrence, maintenance, and circulation of *T. vivax* and *T. evansi,* particularly in areas outside the tsetse belts, may be attributed to other mechanical vectors, such as *Stomoxys* and tabanids [5].

Despite the wide geographic distribution and cosmopolitan nature of *Stomoxys* flies, our knowledge about their blood meal hosts, and the hemopathogens they carry, which will provide insight into vector-host-pathogen interactions, and disease transmission dynamics, is limited and necessitates comprehensive research. Thus, in this study, we aimed to evaluate the role of *Stomoxys* spp. in pathogen transmission by investigating their species diversity, ecological distribution, host-feeding preferences, and the hemopathogens they harbor. We also assessed their vector competence by evaluating their ability to transmit these pathogens. Our findings show that *Stomoxys* flies are competent vectors of multiple pathogens and may contribute to their transmission cycles.

## Results

### *Stomoxys* species diversity and relative abundance are ecology dependent

Diverse *Stomoxys* species were trapped throughout the year. A total of 11,323 adult *Stomoxys* flies were collected from various sites, from the National Reserve, including; Shimba Hills National Reserve and Nguruman Conservancy; to zero grazing ecologies. The flies were identified morphologically using specific keys to the species level according to [23] as *S. calcitrans, S. sitiens, S. niger niger, S. niger bilineatus, S. boueti,* and *S. taeniatus* (Fig 1A). The sampling sites varied in species richness with some having only three species, while others had up to six. *S. calcitrans* was identified in all study sites while *S. taeniatus* was only found in Kajiado County. *S. calcitrans* has a body appearance characterized by three dark spots on each of the second and third segments. *S. sitiens* abdominal segments resemble that of *S. calcitrans,* but the dark spots are more transversely elongated. *S. niger niger* appears to have grey coloration with well-defined and dark stripes on abdominal segments. The dorsal view of *S. boueti* appears to have an indistinct dark abdomen and is much smaller. *S. niger bilineatus* has a brownish appearance, with the abdominal segment having defined the dark stripes in a dorsal view. *S. taeniatus* has a brighter golden brown to almost yellowish color and is larger than the other species (Fig 1A). Cytochrome Oxidase 1 (CO1) DNA sequencing validated our morphological species identification, showing our samples formed a distinct cluster aligning with known DNA sequences (Fig 1B). New CO1 sequences, including *S. boueti,* (GenBank Accession number, PP587243) and *S. taeniatus* (GenBank Accession number PQ203543) that were not previously available, were deposited in the NCBI's GenBank (Appendix A in S1 File).

Kajiado followed by Kwale County recorded the highest number of *Stomoxys* species diversity, six and five species, respectively (Fig 1C). The Shannon diversity index shows the varying levels of *Stomoxys* species diversity (Table A in S1 File) across Kenyan counties with Kwale County having the highest Shannon diversity index of 1.36 and Meru County having the lowest Shannon diversity index of 0.29. Overall, the species distribution showed that *S. calcitrans* were the dominating species accounting for ($n=5,547$, 49%), except in Kajiado and Homabay counties. It was followed by *S. niger niger,* ($n=2,938$, 25.95%), *S. boueti* ($n=1,471$, 12.99%), *S. niger bilineatus* ($n=778$, 6.87%), *S. sitiens* ($n=495$, 4.37%), and finally *S. taeniatus* ($n=94$, 0.83%) (Table B in S1 File). Using one of our sites (Kiambu County) we studied the seasonal dynamics of *Stomoxys* flies. *Stomoxys* flies were caught all year round with seasonal variation (Fig 1D). The abundance of *Stomoxys* increased with the rainfall data, there was an annual rainfall of 674 mm with a monthly average of 56 mm in the study county during the study period.

### *Stomoxys* flies blood meal host network analysis demonstrates *Stomoxys* flies feed on a wide host range

*Stomoxys* flies feed on diverse wild and domestic animals. In total 225 fed *Stomoxys* flies were successfully identified to analyze the *Stomoxys*-host feeding network. Fifteen distinct vertebrate blood-meal hosts were identified, including cattle

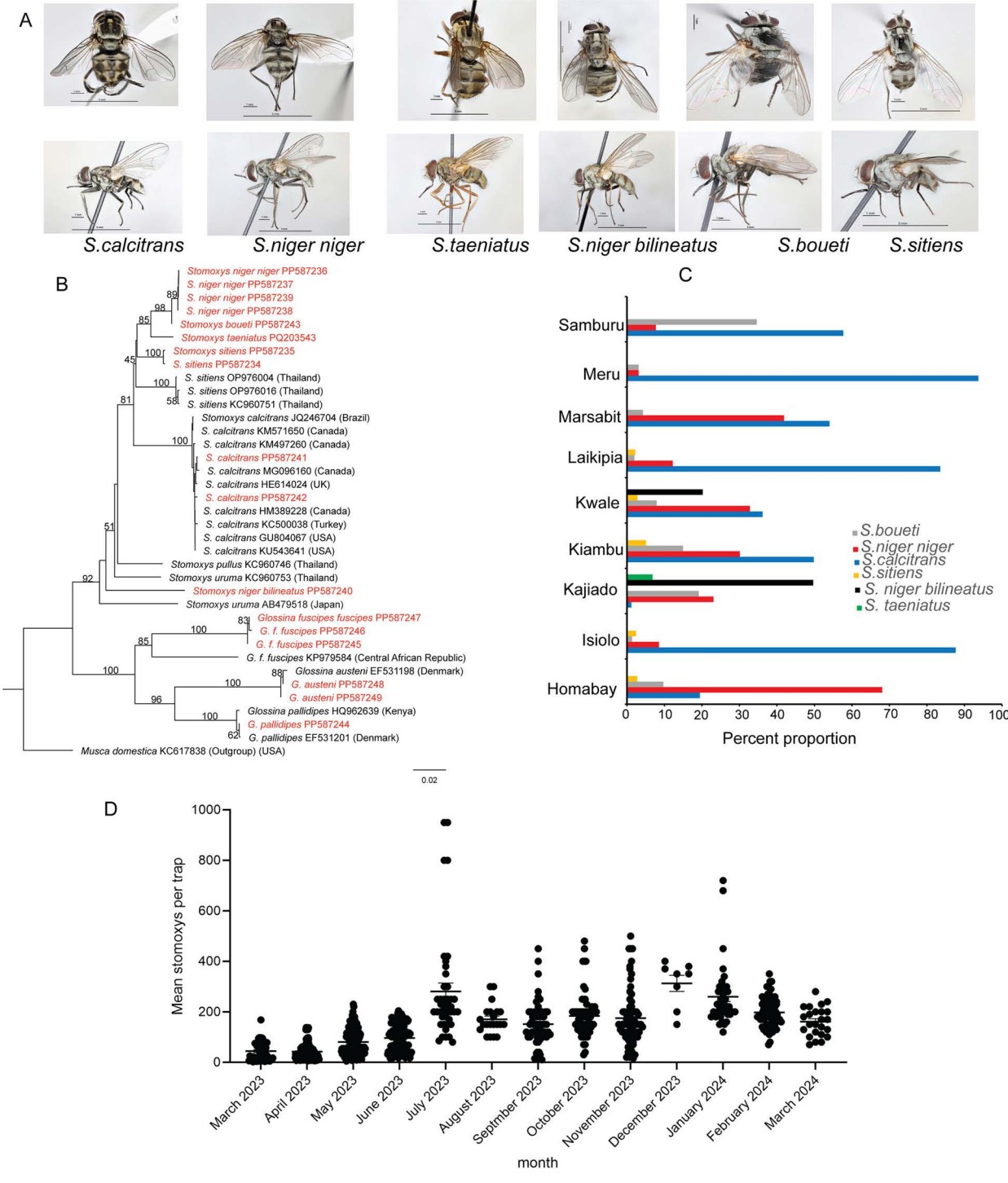

**Fig 1. *Stomoxys* flies morphological identification, molecular characterization, species diversity, and seasonality. (A)** Original image showing the dorsal and lateral view demonstrating the distinct morphological features of the six *Stomoxys* species encountered at various study sites **(B)** Neighbor-joining tree constructed based on aligned sequences of CO1 tree showing the relatedness of the various *Stomoxys* species. Sequences obtained from this study, with their GenBank accession numbers, are highlighted in red. **(C)** The species diversity and their relative abundance in various ecologies. **(D)**. Seasonality of *Stomoxys* at Gatundu site in Kiambu County.

(*Bos taurus*), camel (*Camelus dromedarius*), warthog (*Phacochoerus africanus*), African buffalo (*Syncerus caffer*), goat (*Capra aegagrus hircus)*, waterbuck (*Kobus ellipsiprymnus*) elephant (*Loxodonta africana*), sheep (*Ovis aries*), reticulated giraffe (*Giraffa reticulata*), zebra (*Equus quagga*), baboon (*Papio*), reedbuck (*Redunca redunca*), gazelle (*Gazella gazella*), impala (*Aepyceros melampus*), and human (*Homo sapiens*) (Fig 2A). *S. calcitrans* had the most diverse blood meal hosts followed by *S. boueti* and lastly *S. niger*. Wildlife conservation (Shimba Hills National Reserve) had the most variety of identified blood-meal hosts. Kiambu and Meru counties had the lowest host diversity because of zero grazing in regions where *Stomoxys* were trapped, resulting in a limited number of hosts, mostly only cattle. In general, cattle were the most detected and most preferred host across all species; *S. calcitrans* ($n = 65/225$), *S. boueti* ($n = 13/225$), and *S. niger niger* ($n = 11/225$), (Table C in S1 File). Multiple host feeding was also revealed in some flies where the high-resolution melting (HRM) melt curves revealed two peaks that matched the standard reference (Fig 2B). This was most common in livestock (cattle, sheep, goats, and camels), and was detected once in wildlife (waterbuck and buffalo). This may be because feeding was interrupted before completion.

### *Stomoxys* flies and domestic animals harbor various hemopathogens

Another datum required to elucidate the role of *Stomoxys* for various pathogen transmission dynamics besides blood meal source is to study the pathogens network between *Stomoxys* and some of the most preferred host animals they feed on. Various pathogens were detected in the blood of livestock, which were also common in the *Stomoxys* flies. *Anaplasma* spp., *Theileria* spp., and *Trypanosoma* spp. were shared across all analyzed domestic animal hosts and *Stomoxys* flies. *Coxiella burnetti* was only detected in camels (Fig 3A). *Anaplasma* spp. was the most prevalent pathogen affecting (64.7%) of the camels ($n = 452$). *Trypanosoma* spp. was detected in (12.3%) and *Ehrlichia* spp. in (12.2%) of camels sampled. *C. burnetti* was found in (6%) of the camels. We did not detect any *Theileria*/*Babesia* spp. in camels. Additionally, in cattle ($n = 124$), we found a high prevalence of *Theileria*/*Babesia* spp. (56.6%) and *Anaplasma* spp. (54.1%). Also in cattle, *Trypanosoma* spp. was detected in (10%), and *Ehrlichia* spp., the least prevalent, was found in only (1.6%). A total of 3,451 *Stomoxys* were screened for pathogen diversity across the study counties. Among these, (49.1%) carried *Anaplasma* spp., followed by (19.1%) which harbored *Theileria*/*Babesia* spp., and (9.1%) had *Trypanosoma* spp. For comparison, *Glossina pallidipes* ($n = 1000$) which co-inhabits with *Stomoxys* had pathogen prevalence of *Trypanosoma* spp., *Anaplasma* spp., and *Theileria*/*Babesia* spp., which were detected in (7.5%), (5%), and (11%) of the flies, respectively. To show genetic diversity and evolutionary relationships, a neighbor-joining phylogenetic tree was constructed using the obtained sequences (Appendix B in S1 File), as shown in Fig 3B.

### *Stomoxys* flies are competent mechanical vectors of *T. evansi* and *T. vivax*

*Stomoxys* averagely fed on 9.98 ± 5.5 μL (mean ± SD; $n = 10$ flies) of blood when fully engorged and needed an average of 4.6 ± (2) minutes to fully engorge (Table D in S1 File), number in parenthesis is the standard error of the means. *T. evansi* survived in various tissues of *Stomoxys* after immediate disruption of feeding. About (30%) of *Stomoxys* fed on infected mice showed parasites in the proboscis if feeding was interrupted, for up to five minutes. However, more than (80%) of the flies fed on infected mice had parasites in their crop and gut when feeding was interrupted. *T. evansi* survived up to six hours in the gut of *Stomoxys* which shows a possibility of delayed transmission of trypanosomiasis (Fig 4A). In the first three hours, the trypanosomes were very active swimmers and gradually became inactive after four hours and were all dead six hours post-feeding by flies. We demonstrate that *Stomoxys* flies transmit *T. evansi* through *in vivo* experiments using laboratory mice, with 8.3% (2/24) (Fig 4B) of mice with patent parasitemia detected by microscopy by day seven after infection assays. The wild *T. evansi* strain showed moderate virulence as the mice maintained a peak of parasitemia ($1 \times 10^8$ trypanosomes/mL blood) for several days and died between the 10th and 14th days, respectively with mild clinical symptoms including no appetite to feed, isolation from the group, lethargy, raised fur, and low PCV. We found longer survival times for *T. vivax* (16 hours) in *Stomoxys* guts as compared to *T. evansi* (Fig 4A). Furthermore, we found a 30%

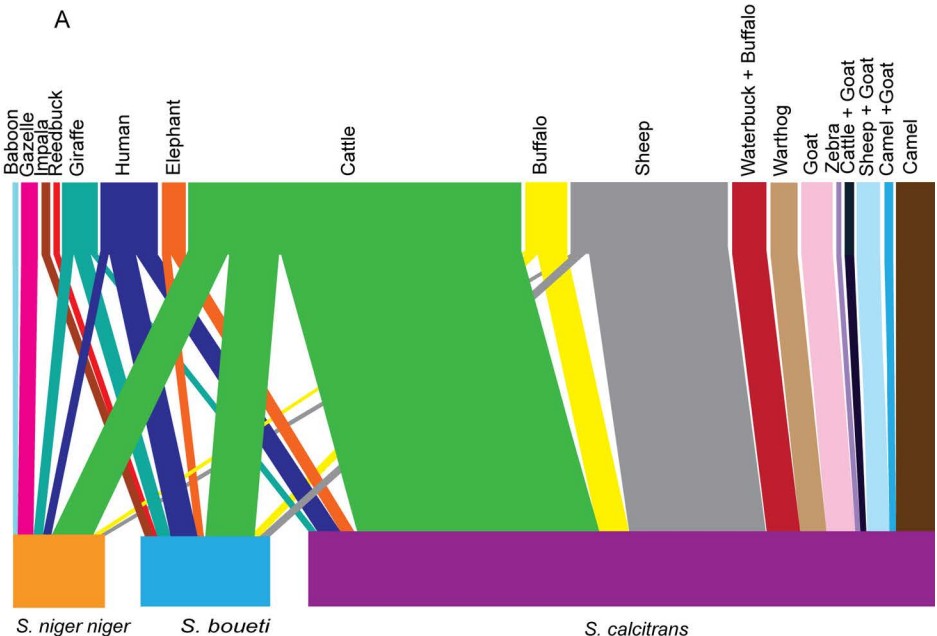

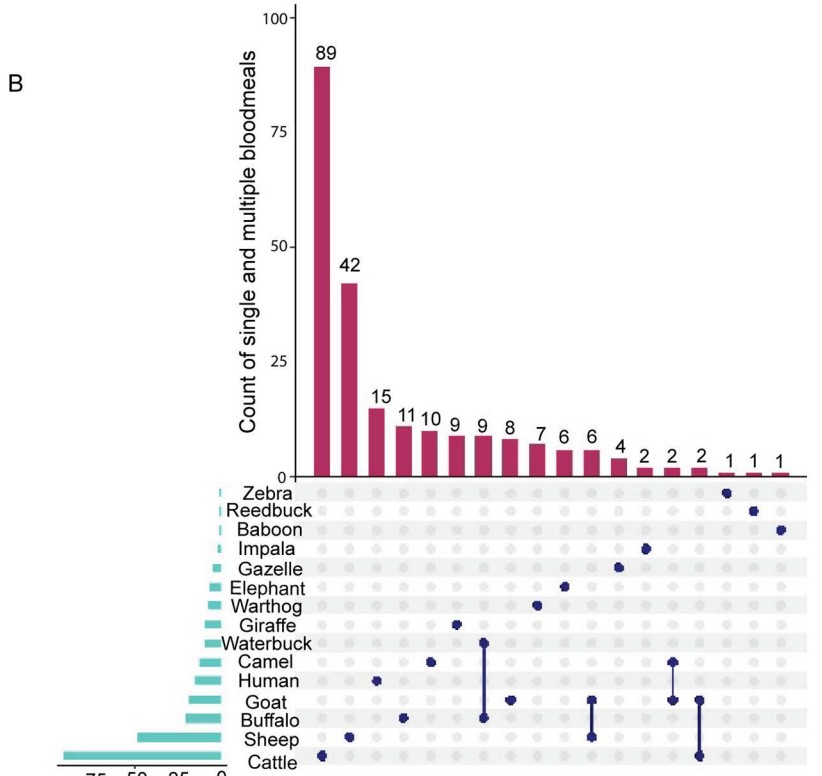

**Fig 2. Identification of vertebrate hosts from blood meal analysis of *Stomoxys* spp.** (A) A *bipartite* network graph showing feeding interactions between hosts and blood-fed *Stomoxys* spp. The top bar indicates hosts while the bottom bar indicates the *Stomoxys* spp. while the lines illustrate the interaction. The thickness of a line corresponds to the number of blood-fed hosts detected in the various *Stomoxys* spp. **(B)** An *Upset* plot showing the total number of hosts fed per species and multiple hosts feeding.

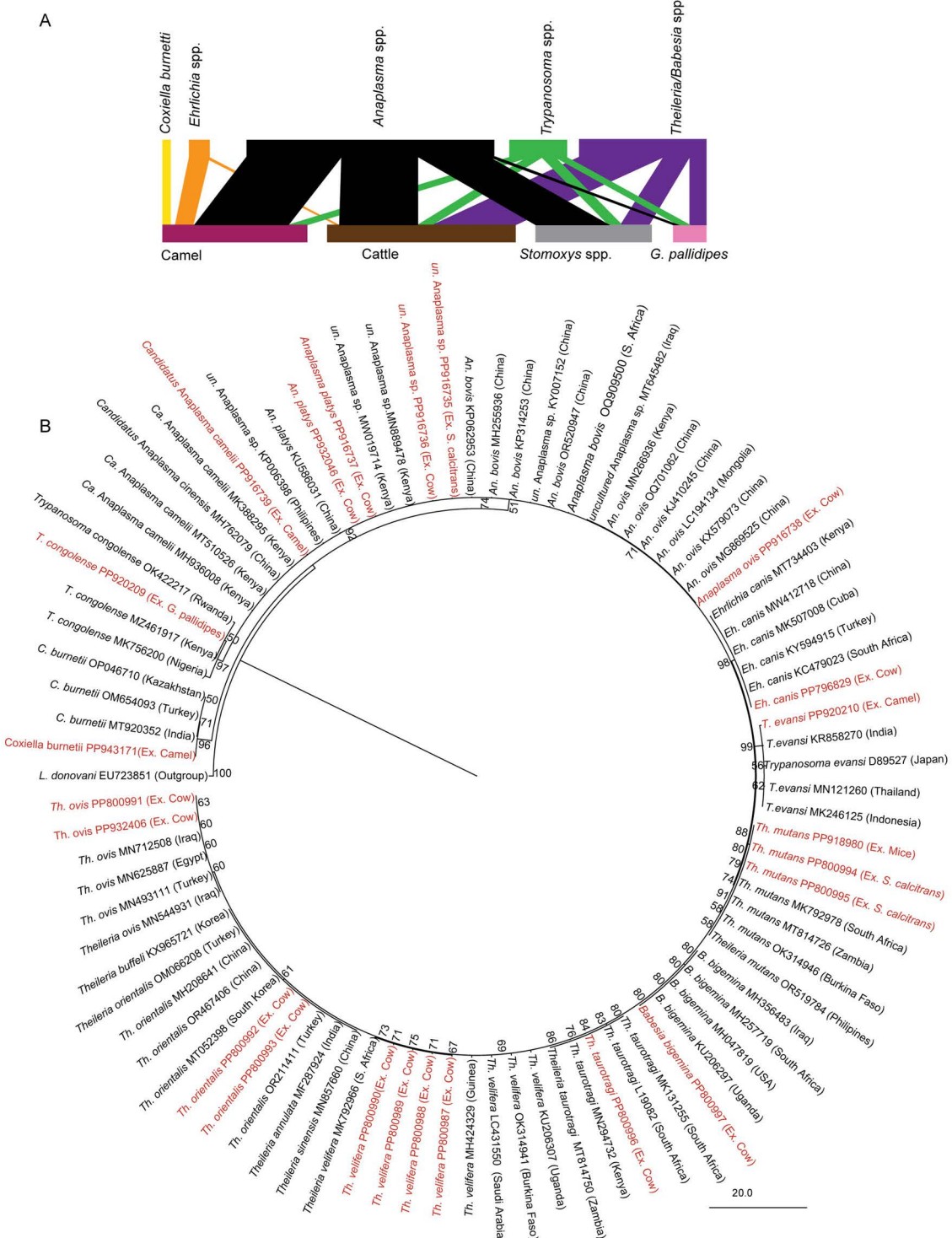

**Fig 3. Pathogen diversity in host animals and vectors from various sites through molecular screening and neighbor-joining tree showing pathogens. (A)** A bipartite network graph showing pathogen interactions between hosts (camel and cattle) and vectors (*Stomoxys* spp. and *Glossina* spp.). Pathogens are indicated by the top bar, while the bottom bar shows their hosts and vectors. The lines illustrate the interaction. The thickness of a line corresponds to the number of pathogens detected in either the hosts or vectors. **(B)** Neighbor-joining tree constructed based on sequences of pathogens from host animals and vectors. Sequences obtained from this study, with their GenBank accession numbers, are highlighted in red.

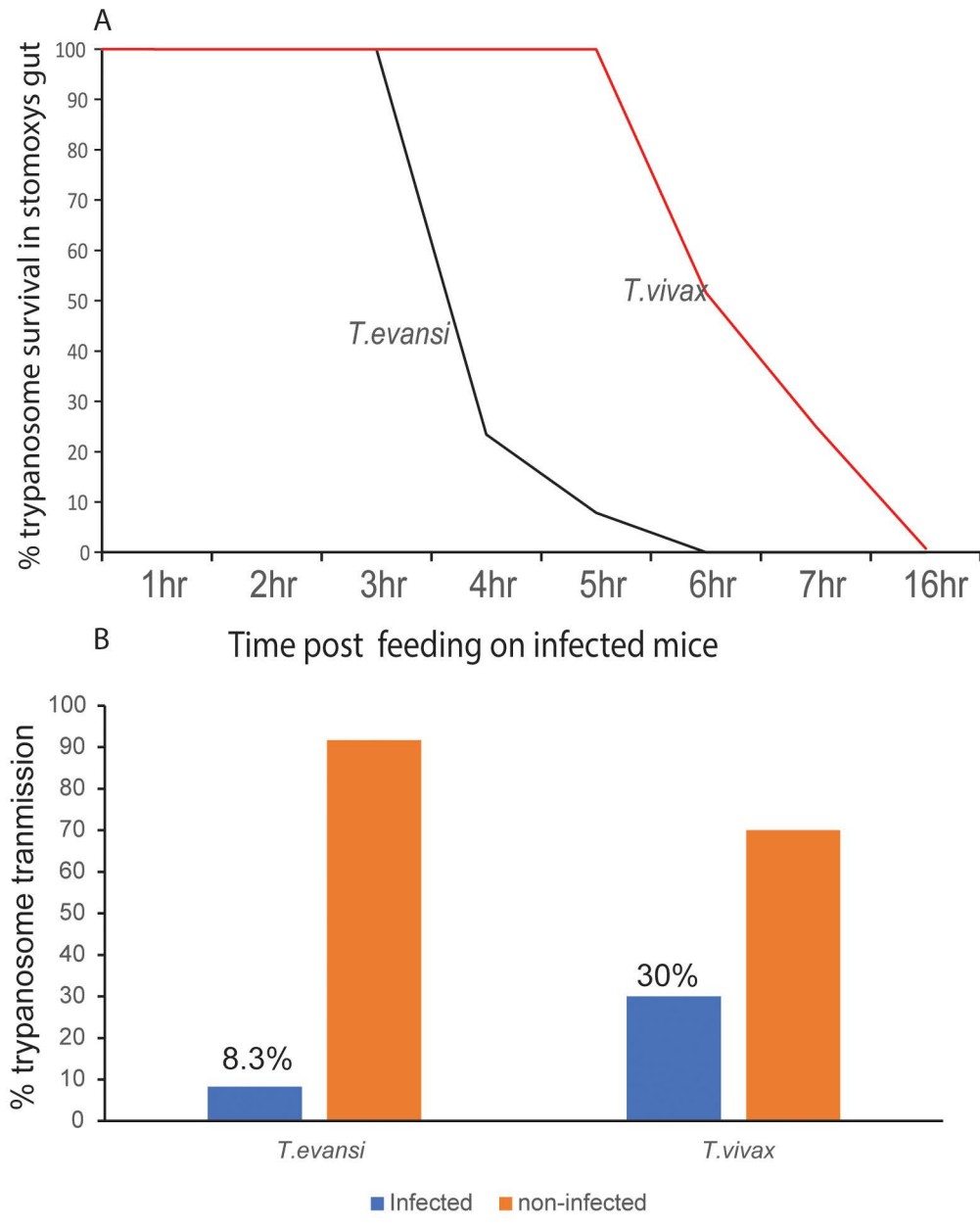

**Fig 4. Vector competence of *Stomoxys* spp. to transmit trypanosomes. (A)**. Graph showing the survival of *T. evansi* and *T. vivax* in *Stomoxys* gut. *T. vivax* exhibits a longer survival time in the gut compared to *T. evansi*, with live *T. vivax* detected up to 16 hours post-feeding, while *T. evansi* was only viable for about six hours. **(B)**. The success of *in vivo* infection. 30% of mice bitten by flies carrying *T. vivax* got infected, compared to only 8.3% for *T. evansi*.

(3/10) higher transmission success rate in *T. vivax* as compared to *T. evansi* (Fig 4B). The incubation period varied from 6 days to 11–34 days in the three *T. vivax*-infected mice, showing individual variation. The *T. vivax* IL 2136 strain exhibited a moderate level of virulence as the infected mice also sustained a high parasitemia of $1 \times 10^8$ trypanosomes/mL blood for several days and died on the 6th, 8th, and 12th days. For comparison, we did five of the same mechanical infection experiments with *G. pallidipes* and all transmitted *T. evansi* to five mice, 100% efficacy. This demonstrated the vector competency variation between *Stomoxys* spp. and *G. pallidipes*.

**Pathogen transmission occurs through feeding bites on experimental mice by field-collected *Stomoxys* flies**

We finally asked if field-collected *Stomoxys* flies can transmit pathogens they harbored by allowing field-trapped *Stomoxys* flies to feed on healthy mice. We demonstrated that wild caught *Stomoxys* flies are capable of delayed transmission of various pathogens they harbored in *in vivo* experiments. *Stomoxys* flies transmitted *T. mutans* (GenBank Accession Number, PP918990) into healthy mice after delayed feeding in the field (Fig 3B) Furthermore, all mice showed *Anaplasma spp.* infection microscopically only. These pathogens had low virulence as the mice showed no clinical symptoms and no mortality of the mice was recorded for ≥ 120 days.

## Discussion

In this study, we aimed to understand *Stomoxys*-host-pathogens network interaction to gain insight into the role of *Stomoxys* flies in disease transmission dynamics, and how transmission networks of pathogens-vectors-host function. Of the 18 globally distributed *Stomoxys* species, 14 are found in Africa [1]. We found a year-round wide distribution of six species of *Stomoxys* including *S. calcitrans, S. sitiens, S. niger niger, S. niger bilineatus*, *S. boueti,* and *S. taeniatus* that varies in abundance and diversity in nine regions including in three tsetse infested ecologies. With our wider geographic coverage, we reported only six species of *Stomoxys* as compared to Mihok et al. [24], who trapped 10 species from Nairobi National Park. *Stomoxys* species complexity varies between ecologies; the national reserve had more species as compared to zero grazing ecologies. For example, species diversity was notably high in Kwale County, most likely due to the availability of numerous breeding sites generated by the forested terrain [25]. Kiambu County, where zero grazing is implemented, had a high population density of mainly three species, which is due to the availability of readily available breeding substrate [26]. Isiolo ($n = 2,514$, 22.20%), Kajiado ($n = 1,373$, 12.12%), and, Homabay ($n = 545$, 4.81%) counties exhibited a considerably high population of *Stomoxys*, which could be attributed to the habitat and semi-arid climate that encourages the growth of the flies [25] Marsabit ($n = 74$, 0.65%) and Samburu ($n = 26$, 0.23%) counties had the least abundance, which could be attributed to the environment, which is a hot and arid climate. This is not friendly to *Stomoxys* because high temperatures have been reported to cause a drop in survival in immature fly populations [27].

To get insight into the role of *Stomoxys* in disease transmission dynamics, we need to understand the natural feeding habits of *Stomoxys* flies from various ecologies. We showed that *Stomoxys* flies feed on a wide range of wild and domestic animals, including humans, which is comparable to tsetse flies [28,29] and corresponds to prior research findings [2,3,30]. From our study, *Stomoxys* need an average of four minutes to complete feeding. This may induce host defense and interrupted feeding, which will result in multiple hosts feeding and pathogen transmission [31]. This disruption of *Stomoxys* feeding before blood meal completion enables the vectors to switch to new hosts to continue feeding, which serves as the basis for the mechanical transmission of pathogens [5]. In general, the relatively diverse host range of feeding suggests that *Stomoxys* may take a more opportunistic approach to host selection, potentially responding to the availability of susceptible hosts in its environment [30]. We found 225 blood-fed *Stomoxys* out of 3,451 showing that most *Stomoxys* flies caught using traps are often seeking hosts for a blood meal, which is also true for other hematophagous insects [32]. Moreover, blood digestion starts more rapidly in *Stomoxys* as compared to other hematophagous flies [33]. Thus, the low rate of blood meal identifications could be explained by the degradation of host DNA during digestion in the fly midgut. Furthermore, as we demonstrated, *Stomoxys* ingests a minimal quantity of blood into its midgut; even while feeding on laboratory-restrained mice, it takes only 10 μL in four minutes. Nevertheless, the variety of hosts we successfully identified includes diverse wild, and domestic animals and humans. The diversity of blood meals can be due to the flies' high mobility, their opportunistic feeding behavior, and their frequent feeding habit. Furthermore, trap position and ecologies may influence the range of host species *Stomoxys* may feed on. For instance, Mavoungou et al. [34], demonstrated that *Stomoxys* flies sampled in canopies mainly fed on arboreal species. We can also notice the absence of small mammals (e.g., rodents) within the diversity of host vertebrates we identified. This may be explained by the trophic preferences of *Stomoxys* flies, the same as tsetse for large vertebrates [28,35,36].

Such a diverse feeding host will expose *Stomoxys* to diverse pathogens as the host varies in their pathogen reservoir capacity [5], which is shown in our pathogen network result. The epidemiology of animal diseases includes the biting rate of vectors on infected hosts and the probability of vectors feeding on different hosts. These are key parameters for understanding the transmission of these infections. Molecular pathogen screening led to the identification of various pathogens that showed epidemiological overlap and interactions between hosts and vectors in the study area. Concerning the pathogens network, we detected high infection rates of *Anaplasma* spp. both in selected domestic animals and *Stomoxys*. The detection of the pathogen from the biting flies confirms the possibility of these flies acquiring and maintaining these pathogens. Ticks including *Rhipicephalus decolaratus, R. simus, R. microplus*, *R. evertsi*, and *Hyalomma marginatum rufipes,* are among the *Anaplasma* biological vectors [37]. However, according to a report conducted by Oliveira et al., seroprevalence and the presence of tabanids and stable flies are associated with bovine exposure to *A. marginale*, which is widespread in Costa Rican dairy herds [10]. Similarly, Bargul et al. demonstrated *A. camelii* transmission by *Hippobosca camelina*, but the same pathogens were not detected in ticks collected from *A. camelii*-infected camel [38]. Another pathogen found with high prevalence both in the host and *Stomoxys* was *Theileria* spp., and our *in vivo* experiment demonstrated the successful transmission of *T. mutans* by *Stomoxys* flies. Similarly, *Theileria* DNA was detected in stable flies, in the case of *T. orientalis* at least for two hours after blood-feeding in a study done by [39]. Additionally, we detected *C. burnetii*, a pathogen that causes Q fever, only in camels in Northern Kenya. Previous research has established a clear link between camel exposure, seroprevalence in camels, and human Q fever infections [40–42]. Experimentally, ticks have been shown to transmit *C. burnetii* and may serve as reservoirs between outbreaks due to their lengthy lifespan [42]. This could explain why we did not detect any *C. burnetii* DNA in either of the biting flies from the area suggesting that it is a tick-borne pathogen [41,43].

In our *in vivo* experimental studies, we discovered that *T. evansi* could actively persist in several tissues of *Stomoxys* flies. *Stomoxys* flies displayed motile *T. evansi* in the proboscis after immediate feeding disruption, which could be observed for up to five minutes. These findings imply that the mouthparts of *Stomoxys* species do not promote trypanosome survival for long [15]. This may be due to the direct transit of blood to the midgut during eating, which leaves very little blood in the proboscis [44]. Our findings are consistent with those of Sumba et al. [15] who confirmed that motile and presumably viable trypanosomes remained in or on the proboscis for around 5–7 minutes after feeding was terminated. While in the midgut, we established *T. evansi* could survive for up to six hours in the gut of *Stomoxys*. This survival capability of *T. evansi* enables a second possible mechanism of transmission, namely regurgitation [16]. Reports in other literature summarized different survival times for various trypanosome species in different biting flies. For instance, Sumba et al. (1998) [15] found that *T. congolense* could live up to three and a half hours and *T. evansi* up to eight hours in the guts of *S. n. niger* and *S. taeniatus*. Additionally, Getahun et al. [3] found that *T. congolense* could live for three hours and trypanozoons for five hours in the midgut of *S. calcitrans* [3,15,16]. Additionally, *Stomoxys* flies are efficient mechanical vectors of *T. vivax*. We showed that *T. vivax* survived in the *Stomoxys* gut for a longer period as compared to *T. evansi* for unknown reasons. The experimental assay showed that *in vivo* transmission of *T. evansi* and *T. vivax* by *Stomoxys* flies was successful with variable success rates. Our findings align with that of Mihok et al. [16] where it was established *S. calcitrans* transmits multiple trypanosome species with various transmission rates. A contrary finding by [45] using relevant host cattle-*T. vivax*-and *S. calcitrans* interaction reported that *S. calcitrans* could not transmit *T. vivax* to cattle possibly because of the transmission experiment design. The authors released the flies into a pen with both healthy and infected animals, and flies may have been more attracted to an infected host than a healthy one [46]. In our protocol establishment, when both the infected mice and healthy mice were kept together, we found no transmission after 20 trials. Furthermore, the interrupting feeding was done after 1.5 minutes. Also, in our protocol establishment experiment, when flies were allowed to feed for more than 1 minute, and transferred to a new host, they lost motivation to feed immediately and even those that fed later did not transmit. It seems mechanical transmission of trypanosomes is time-sensitive. Furthermore, mechanical transmission is parasitemia-dependent [47]. We found the best parasitemia range for mechanical transmission

was $1 \times 10^8$ trypanosomes/mL of blood. The other factor could be that *Stomoxys* species may vary in their vector competence. In our experiment, we kept the *Stomoxys* species complex intact mainly composed of three species as a matrix.

In conclusion, *Stomoxys* spp. are significant but often overlooked vectors of livestock diseases. Their wider geographic distribution, fast reproduction, species diversity, year-round presence, diverse feeding habits, and the plethora of pathogens harbored ensure a continuous risk of pathogen transmission. Additionally, their successful vectorial capacity of transmitting *T. evansi, T. vivax, Anaplasma* spp., and *T. mutans* as shown by our *in vivo* experiments demonstrate the economic importance of *Stomoxys* flies in livestock disease transmission. *Stomoxys* flies play a key role in the spread and maintenance of *T. evansi* and *T. vivax* in the wide geographic regions of the world. In the future, it is important to do vector competence experiments using a specific *Stomoxys* spp. with relevant host animals-pathogens interaction. In our experiment, we kept the natural species complex, composed of mainly *S. calcitrans S. niger,* and *S. boueti* matrix intact. This means we did not try to separate *Stomoxys* by species, but in the future, it is important to do individual species vector competence.

## Materials and methods

### Ethics statement

This study was conducted in strict adherence to the approved experimental guidelines and procedures set forth by the Animal Care and Use Committee (IACUC) of the International Centre of Insect Physiology and Ecology, icipe (REF No.: icipeACUC2018-003-2023) and the Ethics Review Committee of Pwani University (REF No.: ERC/EXT/002/2020E). Farmers and pastoralists were sensitized about the research study and how the findings could benefit the farmers. Sample collection from livestock was done after obtaining verbal consent from farmers, as most of them (camel herders) could not read or write. Animal experiments complied with the approved guidelines and mice were handled carefully to ensure minimum distress.

### Study sites

Field sampling took place in nine selected counties in Kenya at various times from October 2021 to November 2023. The sampled ecosystems in each county were inhabited by a variety of livestock species, wildlife hosts, and humans. These counties included: Kwale County (4.2572° S, 39.3856° E), Isiolo County (0.3257° N, 38.1961° E), Samburu County (1.7299° N, 37.3079° E), Kiambu County (1.0131° S, 36.9051° E), Kajiado County (1.7617° S, 36.0255° E), Marsabit County (2.4426° N, 37.9785° E) Homabay County (0.4368° S, 34.2060° E), Laikipia (0.2924° N, 36.8985° E), and Meru County (0.0515° N, 37.6456° E) (Fig 5).

### Blood collection and fly trapping

**Blood collection.** Blood collection from a total of 452 camels and 124 cattle was done from October 2021 to November 2023. Our initial data determined the sample size, which indicated an infection rate of 4% in camels and 5.7% in cattle caused by *Rickettsiae* spp. the lowest prevalence among pathogens. We utilized this information as a basis for calculating the sample size following the formula; $n = \frac{ln(\alpha)}{ln(1-\rho)}$ as outlined by [48]. At least 5 mL of blood was drawn from the jugular vein of each animal and collected in vacutainer tubes containing Ethylenediamine tetraacetic acid (EDTA) (Plymouth PLG, UK). The blood samples were initially stored in vacutainers at 4°C until the collection was completed which were transferred to cryovials and stored in liquid nitrogen before being transported back to the Nairobi Duduville campus-*icipe* for molecular identification of pathogens.

**Trapping of flies.** Flies were trapped using five red monoconical traps [49] placed 150 meters apart in nine counties as indicated in (Fig 5). Except for Kiambu County, all trapping sites were located in wide woodland savannah ecologies, away from villages. The selection of these sites was intentional and part of our study design, as we aimed to capture *Stomoxys* across diverse ecological settings to assess their potential role as vectors beyond domesticated environments as most wildlife are known reservoirs of these pathogens. This approach allowed us to investigate their diversity, host

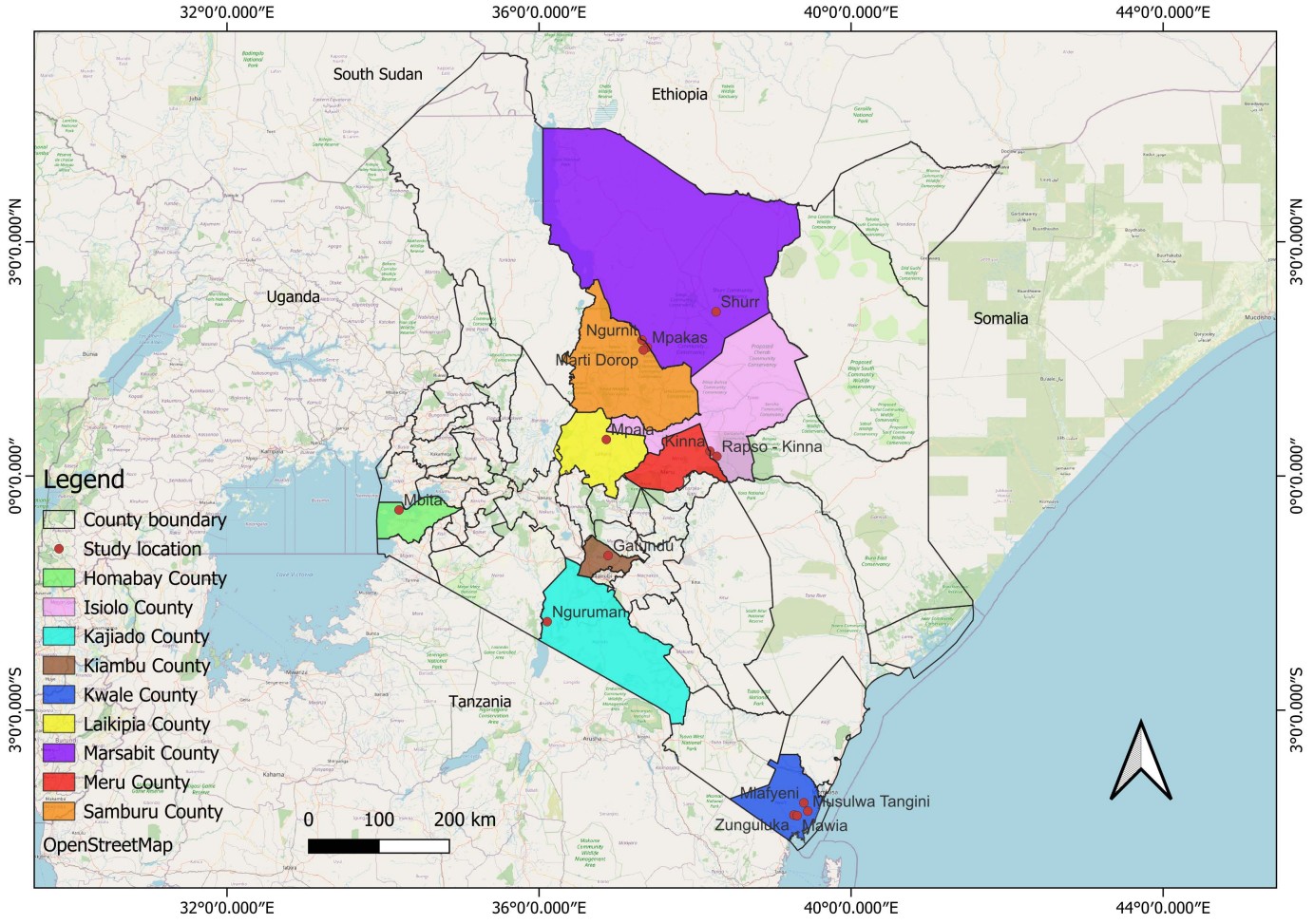

**Fig 5. A map of Kenya showing the sampling sites across the nine selected counties.** This map was created using the open-source software QGIS at 3.34.13 (https://www.qgis.org). The base map from OpenStreetMap is open data, licensed under the Open Data Commons Open Database License (ODbL) available at https://www.openstreetmap.org/copyright by the OpenStreetMap Foundation (OSMF). The direct link to the base layer of the map is found here https://www.openstreetmap.org.

preference, and pathogen carriers in both livestock-associated and wildlife-rich areas, providing a broader perspective on their epidemiological significance, especially in the wildlife-livestock interface. The traps were emptied twice per 24-hour period to avoid flies drying, as needed for clear identification and further pathogens, and blood meal analysis. The flies were later immobilized and preserved in absolute ethanol for further analysis and fresh flies were pinned for morphological identification. Trapping lasted for 5 days.

## Morphological identification of *Stomoxys* spp.

Field-collected *Stomoxys* flies were taxonomically identified to the species level using established taxonomic keys, as outlined by [23]. Briefly, the flies were staged under a dissecting Stemi 2000-C microscope (Zeiss, Oberkochen, Germany) to identify key morphological differences. The head frontal index and the distinctive dorso-abdominal patterns on the second and third segments were key distinguishing features in identifying fly sex and species, respectively. The flies were grouped according to their species and preserved at −20°C for molecular analysis.

## Molecular identification of *Stomoxys* spp. and livestock hemopathogens

**DNA extraction from blood samples and flies.** In the initial stages of DNA pre-extraction, the flies were subjected to a 1% sodium hydroxide immersion for one minute to eliminate any exogenous material on their bodies. Subsequently, they underwent a one-minute rinsing procedure with 1 × phosphate-buffered saline (pH = 7.4). Each fly was mechanically homogenized in sterilized 1.5 mL microfuge tubes containing 750 mg of 2.0-mm yttria-stabilised zirconium (YSZ) oxide beads (Glen Mills, Clifton, NJ, USA) and 80 µL of 1 × PBS using a Mini-Beadbeater-16 (BioSpec, Bartlesville, OK, USA) for one minute. Genomic DNA extraction was done using a DNeasy blood and tissue kit (Qiagen, Hilden, Germany) following the manufacturer's protocol to obtain DNA from flies, cattle, and camel blood for molecular analysis. The quality of the DNA obtained was assessed using a Nanodrop spectrophotometer (Thermo Scientific, Wilmington, DE, United States) by comparing absorbance at 260 and 280 nm and later stored at −20°C for molecular work.

**Molecular identification of *Stomoxys* spp.** For confirmation of the morphological identification of the *Stomoxys* species, we used the extracted genomic DNA in polymerase chain reactions (PCRs) aimed at amplifying fragments of a specific target gene, CO1. The PCRs were conducted in 10-µL reaction volumes, consisting of 5 µl nuclease-free water, 2 µL 5 × HOT FIREPol Blend Master Mix (Solis BioDyne, Tartu, Estonia), 0.5 µL of 10 µM forward and reverse primers (Table E in S1 File), and of 2 µL DNA template. The ProFlex PCR systems thermocycler (Applied Biosystems, Foster City, CA, USA) was used for the amplification process. The amplification conditions were set as described by [50]. PCR amplicons were resolved by electrophoresis under 2% ethidium-bromide-stained agarose gel (100 V for 1 hour), followed by DNA visualization by UV-transillumination (Kodak Gel Logic 200 Imaging System, CA, USA). PCR amplicons were purified using ExoSAP-IT (Affymetrix, Santa Clara, CA, USA) as per the manufacturer's protocol, and outsourced for Sanger-sequencing at Macrogen Inc. (Amsterdam Netherlands).

**Molecular identification of livestock hemopathogens.** We conducted amplification using genus-specific primers for molecular detection of animal blood pathogens of genus *Anaplasma, Theileria/Babesia, Coxiella, Ehrlichia, Rickettsia*, and *Trypanosoma* from the flies and blood. For our vector competence experiments, more species-specific identification of trypanosomes was necessary. We employed PCRs targeting a portion of the internal transcribed spacer (ITS-1) conserved across all trypanosomes. As it is difficult to differentiate based on ITS-1 *T. evansi* and *T. brucei*, we performed *T. brucei*-specific PCR as described in [3]. The PCRs were conducted in 10-µL reaction volumes, consisting of 5 µL nuclease-free water, 2 µL 5 × Blend Master Mix, 0.5 µL of 10 µM forward and reverse primers (Table E in S1 File), and 2 µL DNA template. DNA samples of *T. evansi, T. congolense, T. vivax,* "*Ca*. Anaplasma camelii", *T. parva,* "*Ca*. Ehrlichia regneryi"*, Coxiella burnetti* and *Rickettsia africae* from earlier studies [3,38,43,51] were used as positive controls. For negative controls, 2 µL nuclease-free water was used in place of the DNA template. The amplification conditions were set as described by [50]. PCR amplicons were resolved as described above.

**Phylogenetic analysis.** To get a better understanding of how the various *Stomoxys* species and livestock hemopathogens are similar and different at the genomic level, we performed a phylogenetic tree analysis. The nucleotide sequences acquired in this study were cross-referenced against the known sequences in the GenBank of NCBI nr database (https://www.ncbi.nlm.nih.gov/genbank/). BLAST was used to validate their identity and establish connections with existing deposited sequences [52]. To show the evolutionary relationships, maximum-likelihood phylogenetic trees were constructed using PhyML v. 3.0, employing automatic model selection based on the Akaike information criterion. The tree topologies were estimated through 1000 bootstrap replicates, incorporating nearest-neighbor interchange improvements [53]. The resulting phylogenetic trees were then visualized using FigTree v. 1.4.4 [54].

## PCR-HRM vertebrate blood meal source identification in *Stomoxys* flies

The extracted DNA was used in the analysis of the blood meal source of the fed flies collected in the field. A 10-µL PCR reaction was carried out, consisting of 1 µL of DNA template, 6 µL of nuclease-free water, 2 µL of 5 × HOT FIREpol Eva-Green HRM Mix from Solis BioDyne in Tartu, Estonia, and 0.5 µL of 10 µM of both forward and reverse primers (Table E

in [S1 File]). Positive control vertebrate host samples used as reference include; cattle (*B. taurus*), camel (*C. dromedarius*), donkey (*Equus asinus*), warthog (*P. africanus*), African buffalo (*S. caffer*), goat (*C. aegagrus hircus)*, waterbuck (*K. ellip-siprymnus*) elephant (*L. africana*), sheep (*O. aries*), reticulated giraffe (*G. reticulata*), lesser kudu (*Tragelaphus imberbis*), cheetah (*Acinonyx jubatus*), zebra (*E. quagga*), baboon (*Papio*), gerenuk (*Litocranius walleri*), hartebeest (*Alcelaphus buselaphus*), reedbuck (*R. redunca*), hyena (*Crocuta crocuta*), gazelle (*G. gazella*), impala (*A. melampus*), lion (*Panthera leo*), bongo (*Tragelaphus eurycerus*) and human (*H. sapiens*). The PCR thermal cycling conditions for primer were set as described by [50]. After PCR amplification, HRM analysis was conducted within normalized temperature ranges, from 65°C to 78°C and 88°C to 95°C. The distinct melt curve profiles of the samples were compared against reference standards.

**Experimental infection assays to determine the vector competence of *Stomoxys* to transmit *T. evansi* and *T. vivax***

**Experimental animals.** Swiss White Mice (*Mus musculus*) obtained from *icipe's* Animal Rearing and Quarantine Unit (ARQU) were used for the infection experiment. Both male and female mice used for experiments were about 6–8 weeks old. Each mouse weighed about 24–29g body weight. The mice were housed under normal conditions including a controlled environment with a temperature maintained at 22±2°C, a 12:12 light/dark cycle, and relative humidity set at 50±10%. Each cage housed 5–6 mice and was provided with wood shavings as a bedding material. Their diet primarily comprised commercial pellets (Unga Kenya Ltd) and water, which was provided *ad libitum*. The mice were kept in mice experimental room that was free from biting flies. The experimental mice used in pathogen transmission studies were not immune suppressed.

**Establishment of laboratory colonies of *Stomoxys* spp.** *Stomoxys* spp. of both sexes were trapped from both Gatundu (1.0131° S, 36.9051° E) and around the *icipe*-Duduville campus (1.2921° S, 36.8219° E) and taken to *icipe's* insects rearing units. The mixed species of *Stomoxys* flies were maintained in 75 cm × 60 cm × 45 cm Perspex cages (Astariglas, Indonesia) and fed once daily between 9 am–11 am on warm defibrinated bovine blood obtained from a local slaughterhouse (Choice Meats, Nairobi) and supplemented with 10% glucose and *Parthenium hysterophorus* flowers [55]. The temperature and humidity in the rearing room were kept at 25±1 °C and RH 50±5%, respectively with a 12:12 light/dark photoperiod. Sheep dung was used as an oviposition substrate [56] and developed pupae were picked and transferred to another cage for emergence. The newly emerged flies were used for experimental infection assays.

**Flies feeding efficiency on mice assay.** To ensure our feeding bioassay was functioning properly before we tried the transmission trail, we examined the feeding efficiency of flies on the restrained mouse and quantified the amount of blood taken and the time taken for *Stomoxys* to be fully engorged. We used female *S. niger niger* for this experiment to control species-based variation for the feeding. Flies were individually weighed before and after feeding. The feeding efficiency was calculated by the difference in weight of the individual fly before and after feeding using Sartorius analytic A 120 S capable of measuring 4 decimal places.

**Multiplication of *T. evansi* and *T. vivax* isolates in donor mice.** The strain of *Trypanosoma* used in this study was *T. evansi,* which was isolated from a naturally infected camel (*C. dromedarius*) from Marsabit County, and *T. vivax* IL 2136 which was taken from *icipe's* trypanosomes bio-bank. The stabilates were let to thaw after which parasitemia was checked using microscopy with a × 40 magnification. To ensure the viability of the stabilates before each inoculation, the expected parasitemia was approximately $1 \times 10^3$ to $1 \times 10^4$ trypanosomes/ mL blood depending on the length of storage and initial parasitemia of the blood before storage in the biobank. 200 μL of each stabilate was then inoculated to the mice through the intraperitoneal route.

**Monitoring parasitemia levels in donor mice.** The mice were monitored daily, three days after pathogen inoculation. From our preliminary data, we found that parasitemia was too low to be detected using microscopy before three days post-infection. Briefly, a drop of blood from the snipping of the mouse tail using a blood lancet was placed on a clean slide

and covered using a coverslip as a wet blood smear and examined under a microscope [57]. The parasitemia score was estimated, which correlated to a score sheet, as described by [58]. The period taken from the day post inoculation to the first appearance of trypanosomes in blood was recorded for all mice. This was done until the required parasitemia was achieved ($1 \times 10^8$ trypanosome/mL blood).

**Determination of *T. evansi* and *T. vivax* survival rates in *Stomoxys*.** Given that mechanical transmission of trypanosome parasitemia is known to be dose-dependent [47], our experimental design included the testing of two doses that are typically encountered in natural infection [3] approximately $1 \times 10^8$ trypanosomes/mL blood and $5 \times 10^8$ trypanosomes/mL blood. Following the successful induction of high parasitemia, the next step involved the extraction of whole blood from the donor mouse. This was done by sacrificing the mouse under the standard protocol defined by the Institutional Animal Care and Use Committee (IACUC). Fino-Ject disposable syringe 5 mL/cc with a needle was used for blood collection by cardiac puncture [59] which resulted in 200 µL of blood. The parasitemia was checked microscopically to ensure the blood still had enough parasite concentration. The infected blood was then diluted in clean pre-warmed defibrinated bovine blood collected from the slaughterhouse (Choice Meats, Nairobi) at a 1:1 ratio. The resulting blood mixture, approximately 400 µL, was applied onto clean cotton wool placed in a petri dish. A total of 60 teneral *Stomoxys* flies, which were starved for 24 hours, were placed in a clear acrylic plastic cage with dimensions of $10 \times 10 \times 15$ cm, which was made of a 6-mm-thick Perspex sheet. The flies were fed on the blood-soaked cotton wool that was provided on a petri dish. The infection and spread of trypanosomes within the *Stomoxys* flies were monitored at various time points post-feeding, beginning one hour after the feeding event and continuing at each subsequent hour. To analyze the distribution and prevalence of the parasite within the bodies of the *Stomoxys* flies, we conducted dissections of various body parts, including the mouthparts, crop, and gut. At least five insects per exposure time were examined after immediate interrupted feeding.

**Experimental trials of in vivo transmission of *T. evansi* and *T. vivax*.** Various protocols were tested in the *in vivo* transmission trials for optimization. At first, after the successful induction of high parasitemia within the donor mouse, the donor mouse and recipient mouse were restrained using a restrainer which was made of stainless-steel woven wire mesh with measurements of 0.9 mm per hole and a 400 µM wire diameter. Restrained mice were placed in a $10 \times 10 \times 15$-cm cage made of 6-mm-thick Perspex sheet. Teneral flies n = 20 were introduced and disturbed by the observer to allow them to move from donor to recipient mice. Unfortunately, after more than 20 trials using this experimental method, we did not get any results. We optimized our experiment whereby, once a donor mouse with high parasitemia was achieved, the donor mouse and recipient mouse were restrained using the restrainer described above, and both were placed in separate $10 \times 10 \times 15$-cm Perspex cages. Twenty unfed 1-day-old teneral flies were released in the cage of the donor mouse and allowed to feed for ≤ 1 minute. The timing was done once the proboscis had pierced the mouse's skin to ensure feeding had started and the flies ingested blood from the infected mouse. This was followed by disrupted feeding, where only the fed flies were individually picked using a respirator and transferred to the next cage containing the restrained recipient healthy mouse. Flies were then allowed to complete their blood meal until fully engorged (Fig 6). The flies were subsequently dissected to confirm the presence of parasites in their gut to assess the parasite-feeding success rate. The restrained recipient mouse was then released into the standard mouse cage and monitored daily after three days post-infection.

**Screening for *T. evansi* and *T. vivax* in the recipient mice by PCR.** Three days after infected fly bites, a combination of microscopy and molecular methods was used to confirm the presence of parasites in the blood of potentially infected mice for up to 30 days post-infection. Microscopy was done daily, as described above. Molecular screening was done by collecting blood samples from snipping the mice tails and collecting them in 1.5 ml eppendorf tubes, which contained 80 µL $1 \times$ PBS buffer, pH = 7.4. Blood collection was done after every two days. This was followed by DNA extraction using a DNeasy blood and tissue kit following the manufacturer's protocol. PCR, gel electrophoresis, and sequencing were performed as described above.

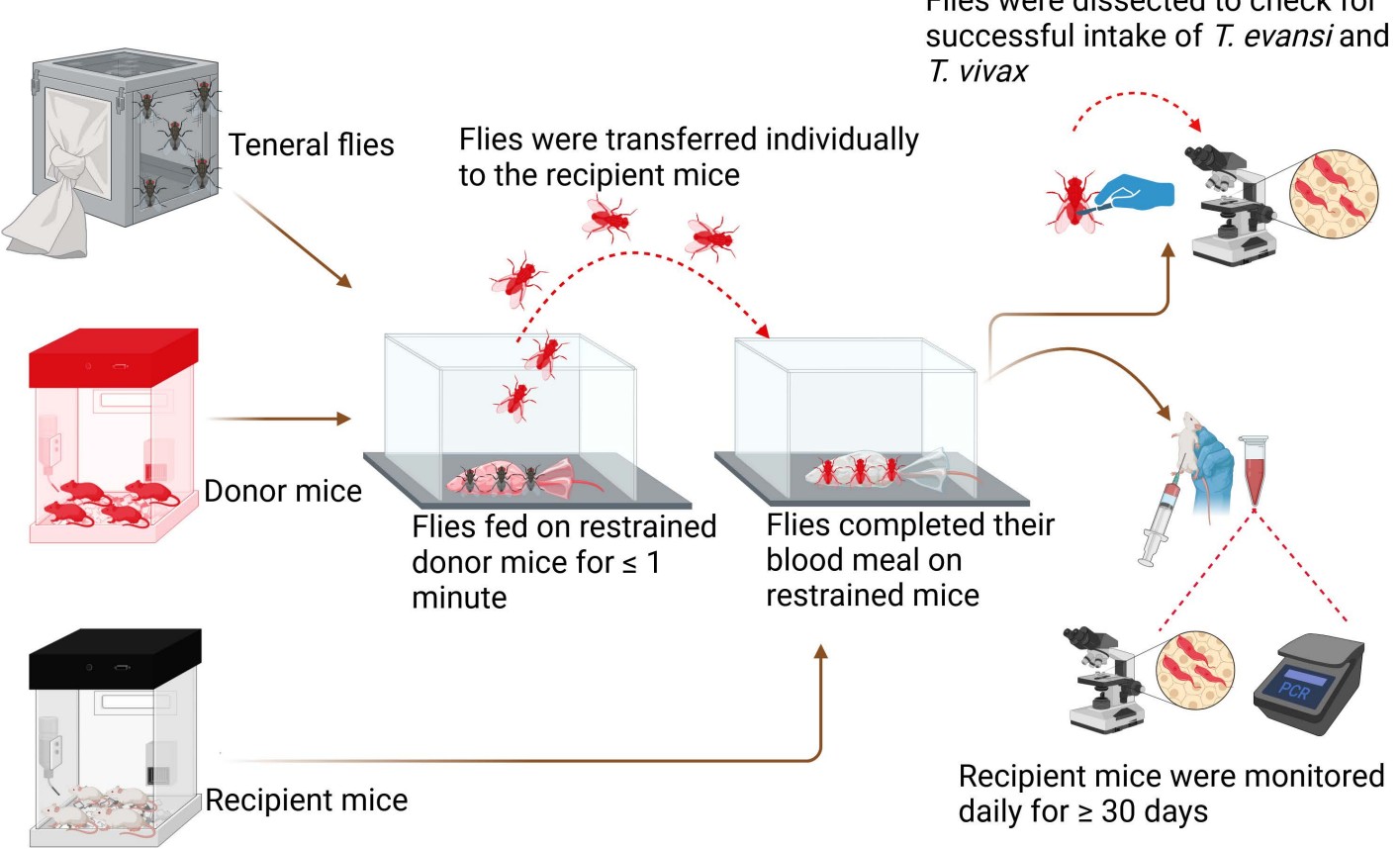

**Fig 6. Laboratory *in vivo* transmission of *T. evansi* and *T. vivax* experimental design.** "Created in BioRender. Getahun, M. (2025) https://BioRender.com/c61b063".

### Determination of vector competence through field bioassay

Fly trapping was done as described above in various study sites. The traps were emptied after 6 hours and the flies were put into 10 × 10 × 15-cm Perspex cages, as described above. The recipient mice were restrained in the restrainer made of stainless-steel woven wire mesh with measurements of 0.9 mm per hole and a 400 μM wire diameter and placed in the cage. The flies were left to feed for 30 minutes before releasing the mice. This was followed by daily evaluation of pathogens in the recipient mice through microscopy and molecular screening, as described above.

### Data analysis

The Shannon diversity index (H) was utilized to define the diversity index of biting flies among study counties and was calculated using R statistical software (R version 4.4.1.). Estimated minimum infection rates (MIRs) of pathogens obtained for the flies were calculated as the number of positive per total number of flies tested ×100. Graphs were visualized using GraphPad software (GraphPad Software, Inc, USA). The *bipartite* R package's interaction network [60] visualized the vectors blood-feeding behavior and pathogen interactions between hosts and vectors, which was generated by R statistical software (R version 4.4.1.). An *Upset* plot displayed the number of flies feeding on specific animal species and those containing blood meals from one or more host species and was plotted using R statistical software (R version 4.4.1.).

Transmission rates of the experimental infection assays were performed by calculating the number of infected mice per total number of transmission trials done ×100.

## Supporting information

**S1 File. Stomoxys Shannon diversity index, Stomoxys density, mammalian host fed on, Stomoxys feeding efficiency, primers and accession numbers.**
(DOCX)

## Acknowledgments

We would like to acknowledge; Dr. Geoffrey Gimonneau and Dr. Mark Desquesnes for useful discussion about infection experiment protocol development; Dr. Steve Mihok for useful discussion and his support in *Stomoxys* identification; James Kabii for his technical support; John Ngiela, Victor Omondi, and Peter Ahuya who helped in the fieldwork; Raphael Mongare and Shadrack Kibet for designing the map of sampling sites; Joseck Esikuri for supplying mice for experimental pathogen transmission assays and Caroline Muya who helped in handling the administrative aspects relating to this study.

## Author contributions

**Conceptualization:** Julia W. Muita, Joel L. Bargul, Daniel K. Masiga, Merid N. Getahun.

**Data curation:** Julia W. Muita, JohnMark O. Makwatta, Ernest M. Ngatia, Simon K. Tawich.

**Formal analysis:** Julia W. Muita, JohnMark O. Makwatta, Ernest M. Ngatia, Simon K. Tawich.

**Funding acquisition:** Daniel K. Masiga, Merid N. Getahun.

**Investigation:** Julia W. Muita, Merid N. Getahun.

**Methodology:** Julia W. Muita, JohnMark O. Makwatta, Ernest M. Ngatia, Simon K. Tawich.

**Project administration:** Merid N. Getahun.

**Resources:** Merid N. Getahun.

**Software:** Merid N. Getahun.

**Supervision:** Joel L. Bargul, Daniel K. Masiga, Merid N. Getahun.

**Validation:** Joel L. Bargul, Daniel K. Masiga, Merid N. Getahun.

**Visualization:** Julia W. Muita, JohnMark O. Makwatta.

**Writing – original draft:** Julia W. Muita, Merid N. Getahun.

**Writing – review & editing:** Julia W. Muita, Joel L. Bargul, JohnMark O. Makwatta, Ernest M. Ngatia, Simon K. Tawich, Daniel K. Masiga, Merid N. Getahun.

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
