## [Decision Letter · Decision Letter 0]

29 Jan 2025

PPATHOGENS-D-24-01930

Stomoxys flies (Diptera, Muscidae) are competent vectors of multiple livestock hemopathogens

PLOS Pathogens

Dear Dr. Getahun,

Thank you for submitting your manuscript to PLOS Pathogens. After careful consideration, we feel that it has merit but does not fully meet PLOS Pathogens's publication criteria as it currently stands. Therefore, we invite you to submit a revised version of the manuscript that addresses the points raised during the review process.

Please submit your revised manuscript within 60 days Mar 30 2025 11:59PM. If you will need more time than this to complete your revisions, please reply to this message or contact the journal office at plospathogens@plos.org. Please include the following items when submitting your revised manuscript:

We look forward to receiving your revised manuscript.

Kind regards,

Junwei J. Zhu, Ph.D.

Guest Editor

PLOS Pathogens

Jeffrey Dvorin

Section Editor

PLOS Pathogens

 Sumita Bhaduri-McIntosh

Editor-in-Chief

PLOS Pathogens

orcid.org/0000-0003-2946-9497

Michael Malim

Editor-in-Chief

PLOS Pathogens

orcid.org/0000-0002-7699-2064

**Journal Requirements:**

At this stage, the following Authors/Authors require contributions: Julia W. Muita, Joel L. Bargul, JohnMark O. Makwatta, Ernest M. Ngatia, Simon K. Tawich, Daniel K. Masiga, and Merid Negash Getahun. Please ensure that the full contributions of each author are acknowledged in the "Add/Edit/Remove Authors" section of our submission form.

- ® on pages: 8 line 156, 10 lines 200 and 207, 12 lines 243 and 256, 13 line 263, and 15 line 291.

Potential Copyright Issues:

i) Please confirm (a) that you are the photographer of 3A, or (b) provide written permission from the photographer to publish the photo(s) under our CC BY 4.0 license.

ii) Figure 2. Please confirm whether you drew the images / clip-art within the figure panels by hand. If you did not draw the images, please provide (a) a link to the source of the images or icons and their license / terms of use; or (b) written permission from the copyright holder to publish the images or icons under our CC BY 4.0 license. Alternatively, you may replace the images with open source alternatives. See these open source resources you may use to replace images / clip-art:

iii) Figure 1. Please (a) provide a direct link to the base layer of the map (i.e., the country or region border shape) and ensure this is also included in the figure legend; and (b) provide a link to the terms of use / license information for the base layer image or shapefile. We cannot publish proprietary or copyrighted maps (e.g. Google Maps, Mapquest) and the terms of use for your map base layer must be compatible with our CC BY 4.0 license.

6) We note that your Data Availability Statement is currently as follows: "All relevant data are in the manuscript and supplementary data. All sequence deposited in NCBI data base". Please confirm at this time whether or not your submission contains all raw data required to replicate the results of your study. Authors must share the “minimal data set” for their submission. PLOS defines the minimal data set to consist of the data required to replicate all study findings reported in the article, as well as related metadata and methods (https://journals.plos.org/plosone/s/data-availability#loc-minimal-data-set-definition).

7) Please amend your detailed Financial Disclosure statement. This is published with the article. It must therefore be completed in full sentences and contain the exact wording you wish to be published.

2) State what role the funders took in the study. If the funders had no role in your study, please state: "The funders had no role in study design, data collection and analysis, decision to publish, or preparation of the manuscript.".

**Comments to the Authors:**

**Please note that one of the reviews is uploaded as an attachment.**

**Reviewers' Comments:**

Reviewer's Responses to Questions

**Part I - Summary**

Reviewer #1: The work describes novel and interesting in vivo experimental procedures, such as mouse transmission of parasitic pathogens. The experimental design is comprehensive and combines field work with molecular techniques and in vivo work. The statistics used is robust and methodology is well described, meriting reproducibility. Interesting insight on Stomoxys distribution feeding habits in Kenya is given. However, the work bears a significant lack of understanding of the parasitological context, has factual mistakes and does not bring any significant novelty.

I would emplore that the work focuses on Stomoxys distribution and behavior in Kenya (I would recommend adding the geographical location to the title of the work). The focus on pathogen transmission and host/vector competence should be generally less focused on, as this part of the work lacks novelty and is experimentally limited.

In general I would recommend a major revision of the works aims and conclusions, as these are unclear and not well delivered.

Reviewer #2: This is a very interesting paper with a novel approach to pathogen transmission in stable flies. The science is good, but the writing is wordy and the paper probably could be trimmed down a bit. The English is good, but sometimes it appears that 2 different people wrote various sections. Most of my comments are merely about housekeeping. for example, when a product is first mentioned, with accompanying source information, the source info does not need to be repeated each timed the product is subsequently mentioned. I made changes in the text for clarity and brevity. There is plenty of information in the text and clarity is important. There are no Key Words. Are they not required? The title is not very specific. Although the results indicated transmission of several hemopathogens, the most specific work was done with 2 species of Typanosomes. A title that read "......competent vectors of Trypanosoma vivax, T. evansi, and multiple livestock hemopathogens" would have searchable hemopathogen names. The stable fly host list of domestic and feral African animals, including humans, is a good addition to the literature. Stomoxys species grouping studies are valuable for future work. Figures are good but hope the published font size will be larger. See other suggested changes in the text.

**Part II – Major Issues: Key Experiments Required for Acceptance**

Reviewer #1: Line 151-164 describe PCR detection of hemopathogens. Did negative controls include uninfected samples to assess the viability of primers, or was only water used? Were positive controls used as well for the selected pathogens? How well do the primers distinguish between different and related species (such as T. evansi, T. vivax and T. brucei)? This should be well assessed and described in the work in order to draw further conclusions from the experimental data.

Lines 217-220 mention checking of parasitemia and administering 200ul to mice. What was real and expected parasitemia (concentration)? This should be added to the methodology. In addition, to assess the vector competence, there should be more focus on the infectious dose required to sustain infection in vivo.

Reviewer #2: There are more than enough experiments discussed in the text to support the results.

**Part III – Minor Issues: Editorial and Data Presentation Modifications**

Reviewer #1: Authors summary: why are tsetse flies mentioned, if they do not transmit T. evansi (in fact T. evansi is not able to be transmitted by Glossina flies, but relies on other vectors, including Stomoxys), and T. vivax is only mechanically transmitted by Glossinas (and by other vectors as well)

African trypanosomes are generally accepted to be T. brucei brucei, T. brucei rhodesiense and T. brucei gambiense. T. evansi and T. vivax are related to them, but they are spread in middle east and America and not generally called “African trypanosomasis”. Generally this should be corrected throughout the manuscript (throughout authors summary, introduction eg. line 72, line 494), better explained and defined, as it is misleading and shows lack of rudimentary knowledge.

Related to this, the spread of T. vivax and T. evansi outside the tse-tse belt is known and explained by range of vectors, including Stomoxys (lines 77-78). This must be better explained in the work.

Line 85: “capacity to become infected” and ability to “transmit hemopathogens” are two absolutely different things. Especially the infection is not explored in this work and therefore this sentence should be modified.

Lines 116-117 have a typological error in the abbreviation of EDTA

Lines 127-128 Why were the trapping sites for insects set out of close proximity to villages, if domesticated animals are in focus of the study – would it be better to carry out the sampling near livestock stables, where the transmission of blood-borne pathogens can be more common and Stomoxys flies (stable dwelling flies) would be present?

Line 129 – methodology for morphological identification of Stomoxys flies – how long did it approximately take from collection to storing the flies at -20°C? Prolonged room-temperature storage could negatively effect the consecutive molecular detection of pathogens.

Line 149: comparing absorbace at 260 and 280 nm is used to assess DNA purity, not concentration, this should be corrected or better explained

Line 199“normal” conditions of mouse housing should be fully described (temperature, night-day cycle, humidity, perhaps even number of animals per cage might be relevant).

Line 221-228 Why was the parasitemia not evaluated during the first three days after infection – was there too low parasitemia expected at early stages of the infection?

Lines 316-317 describe the distribution of various Stomoxys species – this is interesting data and should be explored further – in figure 3 C, I would welcome a graphical representation of the total number of flies identified, not only percentages.

Line 411 describes that “Stomoxys feeds about 9.98 ± (5.5) mL blood”. I hope that this is a typological error and should be microliters and not mililiters. In addition, this statement should be supported by citation.

Line 424 mentions clinical symptoms in mice, these should be described.

Figure 6 legend should be described in more detail

Line 483-484 Mentions strained animals, what does this mean, should it be “starved” instead? Also blood meal is given in mg instead of ml, in contrast to previous mention in the text. This should be also supported by citation

Lines 514-516 mention lack of detection of Ehrlichia and Rickettsia – I find this confusing. Were the PCR primers used to detect the pathogens in hosts, but failed to detect them in the pathogen?

Lines 546-547 sentence is missing a full stop.

Lines 556-557 in conclusion state that “Stomoxys flies may play a significant role in the spread and maintenance of T. evansi and T. vivax in the wide geographic regions of the world.” This is a known fact

Linguistic concerns: the whole text should be re-checked for grammatical coherence and mistakes. Some minor changes include line 198: live body weight should be “body weight” only, mouse age should be precise range etc. Line 412 states “[needs] minutes to fully engorged” – missing “be”. Similar mistakes are present throughout the text and should be found and corrected by the authors.

Reviewer #2: Minor issues refer mainly to writing brevity and clarity. Data are clear as is. Writing changes can be found throughout.

PLOS authors have the option to publish the peer review history of their article (what does this mean? ). If published, this will include your full peer review and any attached files.

**Do you want your identity to be public for this peer review?** For information about this choice, including consent withdrawal, please see our Privacy Policy .

Reviewer #1: No

Reviewer #2: No

**Figure resubmission:**
---

## [Decision Letter · Decision Letter 1]

15 Apr 2025

Dear Dr Getahun,

We are pleased to inform you that your manuscript 'Stomoxys flies (Diptera, Muscidae) are competent vectors of Trypanosoma evansi, Trypanosoma vivax, and other livestock hemopathogens' has been provisionally accepted for publication in PLOS Pathogens.

Best regards,

Junwei J. Zhu, Ph.D.

Guest Editor

PLOS Pathogens

Jeffrey Dvorin

Section Editor

PLOS Pathogens

Sumita Bhaduri-McIntosh

Editor-in-Chief

PLOS Pathogens

orcid.org/0000-0003-2946-9497

Michael Malim

Editor-in-Chief

PLOS Pathogens

orcid.org/0000-0002-7699-2064

Reviewer Comments (if any, and for reference):

Reviewer's Responses to Questions

**Part I - Summary**

Reviewer #1: I would like to congratulate the authors for the great work in reviewing and resubmitting the manuscript. The changes and comments answer all my questions and address my concerns.

After two minor suggested changes I recommend the work for acceptance and publication by PLOS Pathogens

Reviewer #3: The authors have suitably addressed all reviewers comments. The paper is addressing a key area and the results are important for the scientific community and provide key information for future research directions.

**Part II – Major Issues: Key Experiments Required for Acceptance**

Reviewer #1: (No Response)

Reviewer #3: None

**Part III – Minor Issues: Editorial and Data Presentation Modifications**

Reviewer #1: I have two minor changes to suggest:

In the funding (line 689-690) the following sentence is added erroneously: "the specific restricted project donor (written out in full) and grant number;"

Figure 4 lacks in-picture letters (A, B) to correspond to the legend.

Reviewer #3: None

PLOS authors have the option to publish the peer review history of their article (what does this mean? ). If published, this will include your full peer review and any attached files.

**Do you want your identity to be public for this peer review?** For information about this choice, including consent withdrawal, please see our Privacy Policy .

Reviewer #1: No

Reviewer #3: No

---

## [Editor Report · Acceptance letter]

Dear Dr Getahun,

We are delighted to inform you that your manuscript, "Stomoxys flies (Diptera, Muscidae) are competent vectors of Trypanosoma evansi, Trypanosoma vivax, and other livestock hemopathogens," has been formally accepted for publication in PLOS Pathogens.

Best regards,

Sumita Bhaduri-McIntosh

Editor-in-Chief

PLOS Pathogens

orcid.org/0000-0003-2946-9497

Michael Malim

Editor-in-Chief

PLOS Pathogens

orcid.org/0000-0002-7699-2064